# Linking the Low-Density Lipoprotein-Cholesterol (LDL) Level to Arsenic Acid, Dimethylarsinic, and Monomethylarsonic: Results from a National Population-Based Study from the NHANES, 2003–2020

**DOI:** 10.3390/nu14193993

**Published:** 2022-09-26

**Authors:** Can Qu, Ruixue Huang

**Affiliations:** Department of Occupational and Environmental Health, Xiangya School of Public Health, Central South University, Changsha 410078, China

**Keywords:** arsenic, arsenic species, low-density lipoprotein-cholesterol, NHANES

## Abstract

Arsenic (As) contamination is a global public health problem. Elevated total cholesterol (TC) and low-density lipoprotein-cholesterol (LDL-C) are risk factors for cardiovascular diseases, but data on the association of urinary arsenic species’ level and LDL-C are limited. We performed an association analysis based on urinary arsenic species and blood TC and LDL-C in US adults. **Methods:** Urinary arsenic, arsenic acid (AA), dimethylarsinic (DMA), monomethylarsonic (MMA), TC, LDL-C, and other key covariates were obtained from the available National Health and Nutrition Examination Survey (NHANES) data from 2003 to 2020. Multiple linear regression analysis and generalized linear model are used to analyze linear and nonlinear relationships, respectively. **Results:** In total, 6633 adults aged 20 years were enrolled into the analysis. The median total urinary arsenic level was 7.86 µg/L. A positive association of urinary arsenic concentration quartiles was observed with TC (β: 2.42 95% CI 1.48, 3.36). The OR for TC of participants in the 80th versus 20th percentiles of urinary total arsenic was 1.34 (95% CI 1.13, 1.59). The OR for LDL-C of participants in the 80th versus 20th percentiles of urinary total arsenic was 1.36 (95% CI 1.15, 1.62). For speciated arsenics analysis, the OR for arsenic acid and TC was 1.35 (95% CI 1.02, 1.79), whereas the OR for DMA and LDL-L was 1.20 (95% CI 1.03, 1.41), and the OR for MMA and LDL-L was 1.30 (95% CI 1.11, 1.52). **Conclusions:** Urinary arsenic and arsenic species were positively associated with increased LDL-C concentration. Prevention of exposure to arsenic and arsenic species maybe helpful for the control of TC and LDL-C level in adults.

## 1. Introduction

Among the spectrum of lipoproteins, there is no doubt that total cholesterol (TC) and low-density lipoprotein cholesterol (LDL-C) concentrations are critically important affecting factors for the risk of cardiovascular diseases including Alzheimer’s disease (AD) [1], diabetes [2], or hypertension or Parkinson disease (PD) [3]. Cholesterol homeostasis is vital for proper cellular and systemic functions. Cholesterol can contribute to maintain or alter their conformations by binding multitudinous transmembrane proteins. Cholesterol has interactions with masses of sterol transport proteins to promote cholesterol trafficking and adjust its subcellular distribution [4,5]. When cholesterol balance is disrupted, it can not only cause CVDs, but may also be associated with cancer and neurodegenerative diseases [6]. The potential clinical significance of TC and LDL-C have been widely explored as predictive biomarkers or risk factors. For instance, a 10% reduction in LDL-C was associated with a 25% reduction in coronary artery disease [7]. Emerging evidence have raised that environmental pollutions are typically associated with cardiovascular diseases development, therefore, it is needed to investigate the association of environmental pollutants and the TC and LDL-C to provide more evidence for prevention and control of cardiovascular diseases.

Arsenic is a kind of common poisonous contaminant which is widely distributed in nature and can enter human body through food and water, threatening public health. Arsenic can cause a variety of diseases after entering the human body, such as stroke, peripheral vascular diseases and diabetes, and even increase the risk of bladder cancer, kidney cancer, liver tumors, and lung cancer [8,9]. A data analysis using NHANES showed an association between arsenic exposure and insulin resistance [10]. Chronic and high arsenic exposure through drinking water among adults has been correlated with increasing CVDs risk in several studies [11,12]. Monomethylarsonic acid (MMA) is an arsenic form which has been methylated from inorganics, and it has higher toxicity in the human body compared to the second methylation dimethylarsinic acid (DMA) [13]. Through epidemiological analysis, higher arsenic levels had significant higher mortality rates [14], the positive association of arsenic exposure and low high density lipoprotein (HDL) cholesterol [15], and higher arsenic levels had significantly higher blood pressure [16] have been demonstrated. Furthermore, there are also reports on arsenic species exposure and cardiovascular diseases. For instance, DMA level was positively associated with obesity and its comorbidities [17], and higher DMA had higher hypertension risk with an OR of 1.03 (0.94–1.14) [18]. A higher urinary DMA would correspond to heart disease mortality [19]. MMA exposure has been reported to be associated with hypertension risk. Another population-based study showed that individuals with greater exposure to arsenic and lower capacity to methylate inorganic arsenic may be at a higher risk to carotid atherosclerosis [20]. These reports indicated that total arsenic and, in particular, arsenic species exposure need further investigation on their roles in the TC and LDC-C relationships.

Here, in this study, we aimed to investigate the association of urinary total arsenic and arsenic species’ levels with TC and LDL-C among adults through, online available, National Health and Nutrition Examination Survey (NHANES) from 2003 to 2020. Some previous studies focused on the relationship between urinary arsenic concentration and TC and HDL-C. This time, we chose to use the NHANES database to mainly explore the correlation between LDL-C and urinary arsenic concentration, and used the analysis of the correlation between TC and urinary arsenic concentration to support the previous findings. This is a validation and supplement of previous studies, and can make this analysis more targeted. Previous studies based on NHANES are mainly focused on the total arsenic level and cardiovascular mortality rates, thus, it is necessary to assess associations of arsenic species with the LDL-C level among American adults. The exposure assessment includes urinary total arsenic, arsenic acid, DMA, and MMA levels. The outcomes’ values include TC and LDL-C.

## 2. Methods

### 2.1. Study Design

NHANES is a complex multistage sampling design to obtain a representative sample of the civilian, non-institutionalized US population conducted by the US National Center for Health Statistics (NCHS; Centers for Disease Control and Prevention [CDC], Atlanta, GA, USA) [18]. This national cross-sectional study measured urinary total arsenic, six urinary speciated arsenics, and serum TC levels and LDL-C concentrations from participants in the NHANES 2003–2020. NHANES has a survey cycle every two years, except for data collected from 2017 to March 2020. Data collected from 2019 to March 2020 were combined with data from the NHANES 2017–2018 cycle to form a nationally representative sample of NHANES 2017–March 2020 pre-pandemic data. 

### 2.2. Study Population

Participants 20 years of age or older (N = 48,353) from the NHANES 2003–March 2020 were recruited for this urine arsenic and TC or LDL-C association analysis. We excluded 33,449 participants whose total urine arsenic were missing, and 8333 participants were excluded due to missing TC and LDL-C. After excluding other covariates such as marriage and smoking, a total of 6633 participants were included in the study (Figure 1). All participants’ data collection procedures and research protocols were approved by the National Center for Health Statistics Research Ethics Review Board.

### 2.3. Urine Arsenic Assessment

Spot urine samples for arsenic analysis were collected in arsenic-free containers on dry ice, frozen, and stored in the laboratory at −70 °C, and analyzed within 3 weeks of collection [21]. According NHANES Lab Procedures Manual, the coordinator, assistant coordinator, or any MEC staff member was responsible for transporting the urine specimen to the laboratory. Upon arriving at the lab, the Document Urine Collection system verified urine sample information and excluded urine samples that did not meet the testing requirements. Total arsenic exposure was found by analyzing urine through the use of inductively coupled-plasma dynamic reaction cell-mass spectrometry (ICP-DRC-MS), which minimizes or eliminates much of the argon-based polyatomic interference. Diluted urine samples were converted into an aerosol by using a nebulizer inserted within a spray chamber. Aerosols atomize and ionize under thermal energy, the ions and the argon enter the mass spectrometer through an interface that separates the ICP, which was operating at atmospheric pressure (approximately 760 torr), from the mass spectrometer, which was operating at approximately 10^−5^ torr. The mass spectrometer permits detection of ions at each mass-to-charge ratio in rapid sequence, which allows for the determination of individual isotopes of an element. Once inside the mass spectrometer, the ions passed through the ion optics, then through DRC™, and, finally, through the mass-analyzing quadrupole before being detected as they strike the surface of the detector. The ion optics uses an electrical field to focus the ion beam into the DRC™. The DRC™ component is pressurized with an appropriate reaction gas and contains a quadrupole. In the DRC™, elimination or reduction of argon-based polyatomic interferences takes place through the interaction of the reaction gas with the interfering polyatomic species in the incoming ion beam. The quadrupole in the DRC™ allows elimination of unwanted reaction by-products that would otherwise react to form new interferences. Electrical signals, resulting from the detection of the ions, are processed into digital information that is used to indicate the intensity of the ions, and subsequently the concentration of the element. All laboratory tests were partially repeated 2–3 times.

The concentration of speciated arsenics was determined by using high performance liquid chromatography (HPLC) to separate the species coupled to an ICP-DRC-MS to detect the arsenic species. The rest assay of speciated arsenics testing method was the same as the total arsenic determination. In this study, 6 species of arsenic were included in the analysis, namely, urinary arsenous acid, urinary arsenic, urinary arsenobetaine, urinary arsenocholine, urinary dimethylarsonic acid, and urinary monomethylarsonic acid. CV after repeated measurements, including arsenobetaine acid, arsenocholine, dimethylarsinic acid, monomethylarsonic acid, and arsenous acid, respectively, was 3.9–6.1%, 4.1–6.0%, 3.1–7.1%, 2.4–6.6%, 3.8–6.3%, and 3.2–4.8%.

### 2.4. Total Cholesterol (TC) and Low-Density Lipoprotein Cholesterol (LDL-C)

Blood specimens were stored under appropriate frozen (−30 °C) conditions until they were shipped to University of Minnesota for testing. Moreover, according to the NHANES Lab Procedures Manual, quality control has been carried out on the collection, transport, and inclusion of blood samples, and the data entry has been limited by the number of valid ranges. An enzymatic approach specific to cholesterol was used for TC measuring, and esterified cholesterol was converted to cholesterol using enzymatic methods for measurement. The resulting cholesterol was then acted upon by cholesterol oxidase to produce cholest-4-en-3-one and hydrogen peroxide. The hydrogen peroxide was then reacted with 4-aminophenazone in the presence of peroxidase to produce a colored product that was measured at 505 nm. TC analyses were conducted in duplicate with a coefficient of variation (CV) of 1.2–1.3%. The HDL-cholesterol was acted upon by PEG-cholesterol oxidase, and the hydrogen peroxide produced from this reaction combined with 4-amino-antipyrine and HSDA under the action of peroxidase to form a purple/blue pigment that was measured photometrically at 600 nm (secondary wavelength = 700 nm). This is an endpoint reaction that is specific for HDL-cholesterol. HDL analyses were tested in duplicate with a (CV) of 2–3.5%. Free glycerol was converted to glycerol-3-phosphate (G3P) by glycerol kinase. After a series of enzymatic reactions, the absorbance values at 505nm were measured separately. This kind of method is a two-reagent, endpoint reaction that is specific for triglycerides. Triglyceride measuring was conducted in duplicate, and the CV ranged from 1.6 to 2.1%. Serum LDL-cholesterol levels were derived on examinees that were examined in the morning session only. Serum LDL-C was measured only in candidates 12 years of age and older who fasted for at least 8.5 h or more in the morning but not more than 24 h. LDL-C was calculated from measured values of total cholesterol, triglycerides, and HDL-cholesterol according to the Friedewald calculation:[LDL-cholesterol] = [total cholesterol][HDL-cholesterol] − [triglycerides/5]

### 2.5. Other Variables

The questionnaire included questions on sex, age, race, and ethnicity, as well as smoking, drinking, height, and weight. Information on sex, age, race, education, marital status, smoking status, alcohol consumption, diabetes, kidney diseases, and high blood pressure was collected by self-reported questionnaire. Race was self-reported, allowing for multiple categories as Mexican American, other Hispanic, non-Hispanic White, non-Hispanic Black, and other races. Marriages were divided into two categories, with and without partner. In the study, smoking was defined as having at least 100 cigarettes in a lifetime, and alcohol consumption was defined as having at least 12 alcoholic beverages a year. Diabetes diagnosis was divided into three categories: yes, no, and borderline. Body mass index (BMI) was calculated by dividing the weight in kilograms by height in meters squared. Blood pressure levels were measured using a standardized protocol and certified examiners. Serum cotinine was measured by high performance liquid chromatography-atmospheric pressure chemical ionization–tandem mass spectrometry (HPLC-APCI-MS/MS) at the Tobacco and Volatiles Division of the Laboratory Sciences National Center for Environmental Health. Urinary creatinine, used to adjust for urine dilution in field urine samples in statistical models, was measured using a Jaffe rate reaction measured by the CX3 analyzer.

### 2.6. Statistical Analysis

Data analyses were conducted using SPSS (v. 26.0; IBM Corp., Armonk, NY, USA) and R software (v. 4.1.2; R Foundation for Statistical Computing, Vienna, Austria) using the NHANES 2003–2020.3 arsenic analyses sample weights. The distribution of urinary arsenic was right-skewed, and medians and interquartile ranges were used to represent the distribution of urinary arsenic in demographic characteristics. TC and LDL-C were normal distribution. Confidence intervals were set at 95%. The statistical significance level was set at α = 0.05 using two-sided tests. Multiple linear regression models were used to estimate the association of creatinine-adjusted urinary total arsenic quartiles with TC and LDL-C concentrations. Meanwhile, we used generalized linear model to analyze the association between urinary arsenic concentration and TC and LDL-C, and a multifactorial logistic regression was used for risk analysis. Urinary arsenic concentration are quantitative data that conforms to a right skewed distribution, and the multiple linear regression fitted to a direct urinary arsenic concentration is not effective and does not conform to statistical principles, and the results obtained are difficult to interpret. Comparing the extremely high range of urinary arsenic distribution with the extreme low range of urinary arsenic distribution in the population, and exploring that people in which stage of urinary arsenic distribution are more likely to reach the risk range of TC or LDL-C, the odds ratios (OR) of TC, LDL-C, and urine arsenic concentration were calculated at the 20th with 80th percentiles, and 30th with 70th percentiles. To evaluate the non-linear relationship between TC and urinary arsenic concentration, the urinary arsenic concentration was divided into quartiles, the lowest of which was compared with the other three.

Covariates including age, gender, race, BMI, creatinine, cotinine, diabetes, smoking, and drinking, since some covariates such as race lose statistical significance in model fitting, and the overall model fit is not good, in order to ensure that the common covariates remain in the model, the final model abandons some covariates and is subject to the model interpretation below each statistical table. Two covariate-adjusted models were built for multiple linear regression both for TC and LDL-C. The first model was adjusted for BMI, creatinine, and diabetes, while the other adjusted for gender and smoking. Meanwhile, we used two adjusted models to analyze risk for TC and two adjusted models for LDL-C, respectively. In addition, the interaction term between urinary arsenic and participant characteristics was included in two adjusted multifactorial logistic regression models for TC and LDL-C, respectively, and the interaction was deemed statistically significant at *p*-value < 0.05.

## 3. Results

### 3.1. Participant Characteristics

The distribution of sociodemographic and biochemical characteristics of 6633 participants aged 20 years and above (mean age = 50 years) was illustrated in Table 1, of which 51.0% were female and 49.0% were male, 29.5% of the participants had a BMI less than 25 kg/m^2^, 34.1% had a BMI within the scope of 25–30 kg/m^2^, and 36.4% had a BMI beyond 30 kg/m^2^. 

Total cholesterol <200 mg/dL is a component of ideal cardiovascular health according the AHA (American Heart Association) standards [22], and we take it as the cut-off value for the range of healthy TC and dangerous TC. A total of 39.7% of the participants had total cholesterol levels above 200 mg/dL, and the mean level is 192 mg/dL. One study showed that young adults with LDL-C ≥ 100 mg/dL had a higher risk of coronary heart disease [23], and many agree that lower LDL-C is better [24]. Therefore, 100 mg/dL was used as the boundary between the optimal and critical LDL-C concentration in this study. The mean of LDL-C concentration of participants is 113 mg/dL and 62.7% were above 100 mg/dL. The median (interquartile range (IQR)) of concentrations of total urine arsenic was 7.86 (4.07–17.57) μg/L. The urine total arsenic IQR in male was 8.74 (4.55, 18.77) μg/L, which was higher than female with 7.20 (3.70–16.00) μg/L of urine total arsenic IQR. Urine total arsenic concentration distribution varied in races, other Hispanic and other races had higher urine total arsenic concentration. The total arsenic level was higher in men and in participants who had a partner. When TC and LDL-C were divided into the healthy range or the recommended range and the dangerous range, there was no significant difference in total urine arsenic concentration (Table 1).

### 3.2. Median Concentration of Urinary Total Arsenic Varies with Years

The median concentration of total urine arsenic decreased with the increase of investigation cycles, but the detection limit also decreased. The urine total arsenic concentration showed a slight upward trend in the last three survey cycles. The substance concentration below the detection limit was replaced by the lower limit of detection (LLOD) divided by square root of 2 (LLOD/SQRT [2]). The lowest median (interquartile range (IQR)) of concentrations of total urine arsenic was 6.6 (3.42–13.47) μg/L from the survey cycle 2013 to 2014, and the detection limit was 0.26 μg/L (Figure 2).

### 3.3. Association of Urinary Total Arsenic with TC and LDL-C

Table 2 presents the β coefficients and 95% CI that estimated the covariate-adjusted associations of urinary total arsenic with TC and LDL-C. Multiple linear regression analysis was performed on urinary total arsenic concentration divided into quartiles, TC and LDL-C, respectively. The β values of both were statistically significant after adjustment of model 1 and model 2. Especially, a positive association of urinary arsenic concentration quartiles was observed with TC (β: 2.42 95% CI 1.48, 3.36) adjusted for creatine, BMI, and diabetes, and LDL-C (β: 0.95; 95% CI 0.14, 1.77) adjusted for gender and smoking (Table 2). We also further estimated odds ratios of serum lipids by urine arsenic concentrations. The results showed that people with 80% of total urine arsenic levels were more likely to reach the dangerous threshold of TC and LDL-C than those with 20%, and 70 % versus 30 % as well. The OR for TC of participants in the 80th versus 20th percentiles of urinary total arsenic was 1.34 (95% CI 1.13, 1.59) after adjustment for creatine, BMI, and diabetes. (Table 3; model 2). The OR for LDL-C of participants in the 80th versus 20th percentiles of urinary total arsenic was 1.36 (95% CI 1.15,1.62) after adjustment for creatine, BMI, diabetes, age, and drinking (Table 3; model 4).

We divided total urinary arsenic concentrations into quartiles and estimated the ORs for TC and LDL-C; TC and LDL-C were still divided into healthy and dangerous range. As the urinary arsenic concentration increased, the OR of reaching risk threshold for TC and LDL-C both increased in the two adjusted models, respectively (Table 4). The association between total urinary arsenic and TC after adjustment for creatine levels, BMI, and diabetes was consistent in most subgroups, and was somewhat stronger in older, smoking and drinking, as well as LDL-C after adjustment for creatine level, BMI, diabetes, age, and drinking (Figure 3). The relationship between TC, LDL-C, and urinary arsenic concentration was basically the same among gender and education level, but there were differences in some subgroups. In terms of the age group, people aged 40–59 years and ≥60 years with 80% urinary arsenic concentration were more likely to reach a dangerous concentration of TC (OR 95% CI 40–59: 1.37 (1.02, 1.85), ≥60: 1.39 (1.04, 1.82)) or LDL-C (OR 95% CI 40–59:1.39 (1.00, 1.94), ≥ 60: 1.46 (1.09, 1.94)) than those with 20% urinary arsenic concentration. Smokers with urinary arsenic concentration of 80% were more likely to reach TC (OR 95% CI 1.60 (1.24, 2.07)) or LDL-C (OR 95% CI 1.58 (1.22, 2.06)) risk threshold than those with urinary arsenic concentration of 20%, while this association was not significant in non-smokers. Similarly, alcohol drinkers with urinary arsenic concentrations of 80% were more likely to reach TC (OR 95% CI 1.52 (1.25, 1.85)) or LDL-C (OR 95% CI 1.47 (1.21, 1.79)) risk thresholds than those with urinary arsenic concentrations of 20%, whereas this association was not significant in non-smokers.

### 3.4. Six Different Speciated Arsenics were Analyzed

We divided arsenic species concentration into 80th and 20th, TC and LDL-C are still divided into normal and abnormal, then obtained crude ORs with abnormal TC and LDL-C in arsenic species concentrations of 80th versus 20th without correlated factors adjusting. Table 5 shows the unadjusted association between TC, LDL-C and speciated arsenics. Arsenic acid was associated with higher odds of TC risk threshold (OR 95% CI: 1.35 (1.02, 1.79), *p* = 0.037). Dimethylarsinic was associated with higher odds of LDL-L risk threshold (OR 95% CI: 1.20 (1.03, 1.41), *p* < 0.05). Significant association was found between LDL-C (OR 95% CI: 1.30 (1.11, 1.52), *p* = 0.001) and monomethylarsonic.

Table 6 includes the adjusted odds ratio for the association between TC and LDL-C and six speciated arsenics, controlling for various variables in different models. Each urine arsenic species concentration is grouped by 80th and 20th. The remaining analysis methods were consistent with the OR values of TC and LDL-C concentrations in urine speciated arsenic concentration without adjusting for confounding factors. Individuals with dimethylarsinic and monomethylarsonic levels of 80% were more likely to reach the TC and LDL-C risk threshold than those with urinary arsenic levels of 20%. In addition, arsenous acid is associated with higher odds of LDL-L risk threshold (OR 95% CI: 1.23 (1.00, 1.51), *p* < 0.05). Arsenic acid is associated with higher odds of TC risk threshold (OR 95% CI: 1.46 (1.10, 1.95), *p* < 0.05) after adjusting creatine, BMI, diabetes, age, gender, and cotinine.

## 4. Discussion

Arsenic is widely found in food and water and can be excreted in urine after metabolism in body. It is a naturally occurring metalloid derived from the environment, depending on the dose and treat time, arsenic can both be a poison and medicine historically. However, arsenic toxicity could be accumulated by intake from environment, which could be a contributing factor in the development and progression of metabolic disease [25]. Chronic arsenic exposure is a significantly risk factor for pancreatic dysfunction and type 2 diabetes [26]. Areas with high concentrations of drinking arsenic tend to have a high prevalence of hypertension [27]. Arsenic exposure could contribute to the multifactorial origin of metabolic syndrome pathology such as dyslipidemias [28]. In this study, the association between urinary arsenic and TC and LDL-C was studied by using urinary arsenic biomarker assessment of arsenic exposure based on NHANES population survey data from 2003 to March 2020. The results showed that the increase of urinary arsenic concentration was positively correlated with the increase of serum TC and LDL-C concentration. In this cross-sectional study of 6633 adults, a one-quarter increase in total urinary arsenic concentration was associated with a 2.42 mg/L increase in blood TC concentration and a 0.95 mg/L increase in LDL-C concentration. Similarly, as the urinary arsenic quartile level increases, the probability of reaching an unsuitable concentration range for TC or LDL-C increases. The population with urinary arsenic concentration above 70% was more likely to reach the risk threshold of TC and LDL-C, which may increase the risk of cardiovascular diseases. The relationship between urinary arsenic and TC was more significant. People 40 years of age and older, smokers, and drinkers with urinary arsenic levels of 80% were more likely to reach the risk threshold level for TC or LDL-C than those with urinary arsenic levels of 20%. This suggests that the elevated urinary arsenic levels in middle-aged and elderly people, smokers, and drinkers should be of greater concern. Cigarette [29] and alcohol [30] consumption are important factors of cardiovascular and cerebrovascular diseases, and the elderly are also the main groups of cardiovascular and cerebrovascular diseases [31]. The simultaneous exposure of these factors with arsenic may make it easier for blood lipid and lipoprotein indexes such as TC and LDL-C to reach the risk threshold. Among urine speciated arsenic analysis, the dimethylarsinic and monomethylarsonic concentration rising are more likely to related to the abnormal TC or LDL-C. Dimethylarsinic is a major component of urinary arsenic, which may be the main reason for the association between the change of total urinary arsenic concentration and the concentration of TC and LDL-C. The high content of dimethylarsinic is also convenient to detect, which is worthy of more attention in clinical detection. Monomethylarsonic, as one of the metabolites of arsenic in the body, also showed a fair correlation of serum lipids or lipoproteins in this study, which also provided a theoretical premise for the study of the correlation between speciated arsenic concentration and TC and LDL-C concentration and the exploration of the mechanism. It is suggested that serum TC and LDL-C should be examined in patients with elevated urinary arsenic, and that urinary arsenic may be a marker for the diagnosis of cardiovascular diseases.

One study has proved that there is a slight positive correlation between unmethylated inorganic exposure and TC level in 12–17 years old from the 2009–2016 NHANES cycles [32], and the result is consistent with that in our adult population. The TC levels in the rising-high stage of total arsenic exposure after birth was 14% higher than those in the stable-low stage, and LDL-C was 23% [33]. An individual level cross-sectional analysis conducted in India indicated that there is a marginally significant positive relationship between arsenic intake from rice and the changes of LDL (*p*-value = 0.032) [34], which complies with the results in our study. A cohort study of 521 patients followed for 5.01 ± 0.31 years found that compared with the first quartile of the four plasma arsenic concentrations, the OR in the fourth quartile was 1.34 (95% CI: 1.03, 1.75; *p* trend = 0.03), moreover, plasma arsenic concentrations were significantly related with higher risk of dyslipidemia subtypes (including high LDL-C but not TC) [35]. An meta-analysis indicated that arsenic exposure can affect lipid metabolism by increasing serum LDL concentrations and decreasing serum HDL levels [36]. Arsenic had a significant association with lipid profile had been explored in an association analysis between heavy metals such as arsenic and predictive indicators of cardiovascular disease and obesity in children and adolescents [37]. The risk of having atherosclerosis was increased by 5.4-fold (95% CI 2.0–15.0) for one studies subjects with high monomethylarsonic and high homocysteine levels as compared to those with low monomethylarsonic and low homocysteine levels [38], and the LDL concentration is linked to atherosclerosis. There are many studies that suggest arsenic had significant association with TC and LDL [37,39], although some studies have not linked arsenic levels to lipid levels such as TC and LDL-C [40,41]. Therefore, more studies are needed to explore the correlation between TC and LDL-C and arsenic in different populations and at different levels.

Several mechanisms may be involved in arsenic exposure leading to elevated TC and LDL-C. Arsenic is related to the increase of cytokines expression and release of the proinflammatory cytokine, and the response of greatest magnitude corresponds to DMA (III), followed by As (III) [42]. It was found in the experiment that the exposure of arsenic trioxide can cause oxidative stress [43,44]. In a rat model of liver and kidney injury induced by sodium arsenate, TC and LDL-C increased, meanwhile, it showed a significant reduction in activities of antioxidant markers such as superoxide dismutase (SOD) adn catalase (CAT) [45]. Both population and experimental results demonstrate that As promotes LDL oxidation, a key step in vascular inflammation and chronic vascular disease [46]. A study showed that arsenic–protein interactions affect various cellular processes and alter epigenetic regulation, cause endocrine disruption [47]. Some scholars have carried out environmental metal exposure in genetic variants with plasma metabolic patterns in a general population from Spain. The study included a variety of heavy metals such as arsenic, with the correlation analyzed between heavy metal concentration and oxidative stress biomarkers by estimating metabolic principal components (mPC). It did not observe any association between serum arsenic levels with mPCs reflecting other lipoproteins, such as HDL and LDL except VLDL [48]. Consider the mechanism of arsenic exposure and cardiovascular disease, cardiotoxicity caused by As (III) or/and Sb might be concerning disturbing calcium homeostasis [49]. Further studies on these mechanisms are needed to explore the association between arsenic exposure and elevated TC and LDL-C.

There is still room for improvement although considerable preparation needs to be performed before research is carried out to ensure research quality. In this analysis, although some confounding factors related to TC and LDL-C, such as BMI and diabetes, were controlled, some factors, such as whether to take lipid-lowering drugs, were not taken into account due to guaranteeing the sample size, since the missing values are too many. Although TC and LDL-C are typical markers of cardiovascular disease risk, they are not very accurate, and there are still many markers associated with cardiovascular disease events. In addition, this is a cross-sectional study and further population cohort analysis is needed to demonstrate the authenticity of this association.

## 5. Conclusions

Our study found a positive correlation between total arsenic exposure and increased TC and LDL-C concentrations in the adult population. More high-quality population-based cohort studies are needed to assess the role of arsenic exposure in increasing TC and LDL-C concentrations. Our study may provide a basis for early warning of TC and LDL-C concentrations based on arsenic exposure to assess the risk of cardiovascular disease.

## Figures and Tables

**Figure 1 nutrients-14-03993-f001:**
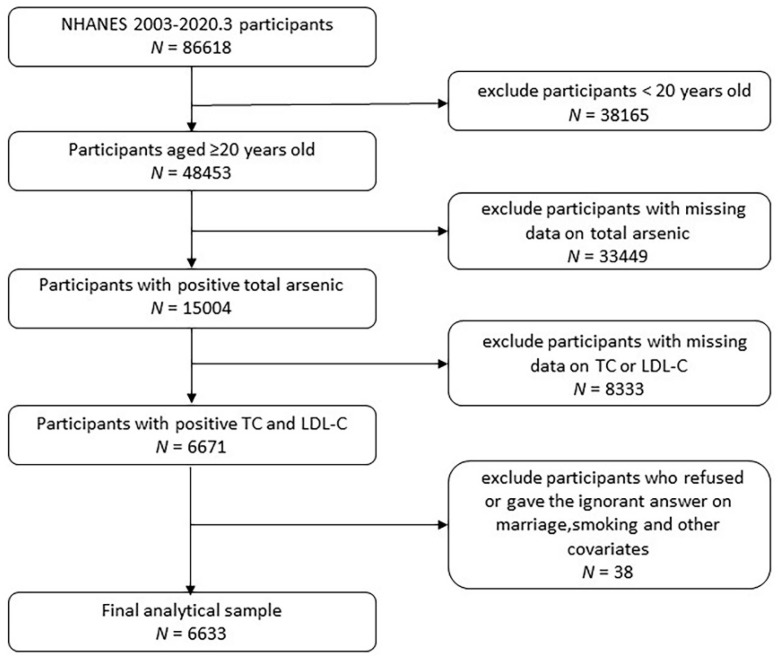
Flowchart of enrolled participants.

**Figure 2 nutrients-14-03993-f002:**
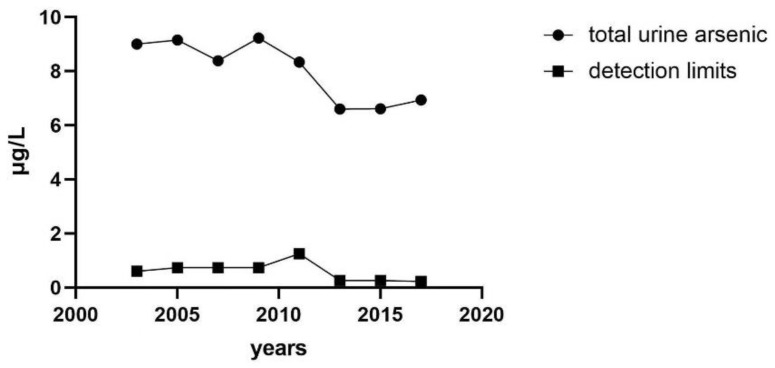
Median concentration of total urinary arsenic varies with years.

**Figure 3 nutrients-14-03993-f003:**
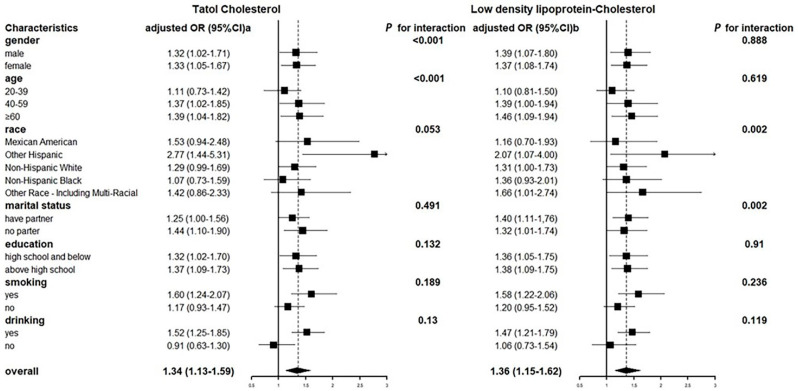
Odds ratios of TC and LDL-C for total urinary arsenic concentrations (μg/L) 80% versus 20% by participant characteristics.

**Table 1 nutrients-14-03993-t001:** Urine arsenic concentrations by participant characteristics.

Characteristics	No. (%) ^a^	Urine Arsenic Median (IQR)-μg/L	*p* Value ^c^
Overall	6633	7.86 (4.07–17.57)	
Sex			<0.001
Male	3247 (49.0)	8.74 (4.55–18.77)	
Female	3386 (51.0)	7.20 (3.70–16.00)	
Age (year)			0.055
20–39	2201 (33.2)	7.53 (4.03–15.84)	
40–59	2138 (32.2)	7.96 (4.02–17.34)	
≥60	2294 (34.6)	8.27 (4.15–18.58)	
Race			<0.001
Mexican American	1071 (16.2)	7.34 (3.99–13.50)	
Other Hispanic	638 (9.6)	10.38 (5.13–20.59)	
Non-Hispanic White	2818 (42.5)	6.54 (3.52–14.04)	
Non-Hispanic Black	1395 (21.0)	8.70 (4.73–22.02)	
Other Races	711 (10.7)	13.13 (5.15–35.12)	
Marital status			<0.001
With partner	4030 (60.8)	8.48 (4.36–18.40)	
Without partner	2603 (39.2)	7.18 (3.74–16.12)	
education			0.004
High school and below	3229 (48.7)	7.47 (3.99–16.20)	
Above high school	3404 (51.3)	8.23 (4.15–18.76)	
Diastolic blood pressure (mmHg)			0.841
<90	6265 (94.5)	7.88 (4.03–17.50)	
≥90	368 (5.5)	7.66 (4.41–18.63)	
Systolic blood pressure (mmHg)			0.835
<140	5516 (83.2)	7.85 (4.08–17.25)	
≥140	1117 (16.8)	7.98 (4.05–19.02)	
BMI (kg/m^2^) ^b^			0.678
<25	1959 (29.5)	8.00 (3.82–19.18)	
25-30	2260 (34.1)	8.00 (4.15–17.64)	
≥30	2414 (36.4)	7.66 (4.21–16.67)	
TC (mg/dL)			0.889
<200	3999 (60.3)	7.87 (4.09–7.23)	
≥200	2634 (39.7)	7.79 (4.06–17.90)	
LDL-C (mg/dL)			0.117
<100	2474 (37.3)	7.64 (3.93–16.94)	
≥100	4159 (62.7)	8.02 (4.14–17.99)	
Creatinine (mg/dL)			<0.001
<80	2187 (33.0)	4.05 (2.20–8.94)	
80–140	2281 (34.4)	8.53 (4.67–18.18)	
>144	2165 (32.6)	11.93 (7.00–26.43)	
Cotinine (ng/dL)			0.001
<0.015	1351 (20.4)	8.12 (4.22–16.80)	
0.015–10.0	3954 (59.6)	8.00 (4.18–18.67)	
>10.0	1328 (20.0)	7.30 (3.77–14.70)	
Hypertension			0.201
Yes	2366 (35.7)	8.09 (4.23–18.00)	
No	4267 (64.3)	7.76 (3.99–17.20)	
Diabetes			0.840
Yes	842 (12.7)	7.72 (4.24–16.24)	
No	5626 (84.8)	7.86 (4.03–17.62)	
Boardline	165 (2.5)	8.17 (4.21–18.32)	
Kidney disease			0.316
Yes	208 (3.1)	7.26 (4.37–16.42)	
no	6425 (96.9)	7.89 (4.07–17.60)	
Smoking			0.007
Yes	3028 (45.7)	7.58 (3.95–16.47)	
No	3605 (54.3)	8.14 (4.18–18.32)	
Drinking			<0.001
Yes	5141 (51.4)	8.10 (4.21–18.00)	
No	1492 (14.9)	7.21 (3.80–15.80)	

IQR, interquartile range. ^a^ Percentage values are weighted. ^b^ Calculated as weight in kilograms divided by height in meters squared. ^c^
*p* values obtained from Wilcoxon rank sum test if comparing between two groups, and obtained from Kruskal–Wallis test if comparing more than two groups.

**Table 2 nutrients-14-03993-t002:** β-coefficients of TC and LDL-C by total urinary arsenic concentration quartiles.

Serum Lipids		β-Coefficients (95% CI ^a^)	*p* Value
TC (mg/dL)	Model 1 ^b^	2.42 (1.48, 3.36)	<0.001
	Model 2 ^c^	2.38 (1.44, 3.31	<0.001
LDL-C (mg/dL)	Model 1 ^b^	0.94 (0.13, 1.76)	0.024
	Model 2 ^c^	0.95 (0.14, 1.77)	0.022

^a^ 95% CI: 95% confidence interval. ^b^ Model 1 adjusted for creatine, BMI, and diabetes. ^c^ Model 2 adjusted for creatine, BMI, diabetes, gender, and smoking.

**Table 3 nutrients-14-03993-t003:** Odds ratios of serum lipids by urine arsenic concentrations.

		70th Versus 30th Percentile		80th Versus 20th Percentile	
Urinary total arsenic (μg/L)		14.40/4.70		22.00/3.49	
TC	Model 1 ^a^	1.17 (1.02, 1.35)	1 [Reference]	1.28 (1.08, 1.52)	1 [Reference]
	Model 2 ^b^	1.21 (1.05, 1.39)	1 [Reference]	1.34 (1.13, 1.59)	1 [Reference]
LDL-C	Model 3 ^c^	1.15 (1.00, 1.32)	1 [Reference]	1.12 (1.03, 1.22)	1 [Reference]
	Model 4 ^d^	1.23 (1.06, 1.41)	1 [Reference]	1.36 (1.15, 1.62)	1 [Reference]

^a^ Model 1 is shown adds odds ratio (95% confidence interval); adjusted for creatine. ^b^ Model 2 is shown adds odds ratio (95% confidence interval); adjusted for creatine, BMI, and diabetes. ^c^ Model 3 is shown adds odds ratio (95% confidence interval); adjusted for creatine. ^d^ Model 4 is shown adds odds ratio (95% confidence interval); adjusted for creatine, BMI, diabetes, age, and drinking.

**Table 4 nutrients-14-03993-t004:** Odds ratios of TC and LDL-C by quartiles of urine arsenic concentrations.

		Quartile 1	Quartile 2	Quartile 3	Quartile 4	*p* Value for Trend
Urinary total arsenic (μg/L)		<4.07	4.07–7.86	7.86–17.57	>17.57	
TC	Model 1 ^a^	1 [reference]	1.14 (0.98, 1.32)	1.13 (0.98, 1.32)	1.24 (1.07, 1.45)	0.020
	Model 2 ^b^	1 [reference]	1.16 (1.00, 1.34)	1.17 (1.00, 1.36)	1.29 (1.11, 1.50)	0.007
LDL-C	Model 3 ^c^	1 [reference]	1.10 (0.95, 1.27)	1.11 (0.95, 1.29)	1.20 (1.03, 1.40)	0.041
	Model 4 ^d^	1 [reference]	1.13 (0.97, 1.31)	1.16 (0.99, 1.35)	1.29 (1.10, 1.51)	0.005

^a^ Model 1 is shown adds odds ratio (95% confidence interval); adjusted for creatine. ^b^ Model 2 is shown adds odds ratio (95% confidence interval); adjusted for creatine, BMI, and diabetes. ^c^ Model 3 is shown adds odds ratio (95% confidence interval); adjusted for creatine. ^d^ Model 4 is shown adds odds ratio (95% confidence interval); adjusted for creatine, BMI, diabetes, age, and drinking.

**Table 5 nutrients-14-03993-t005:** Unadjusted odds ratios (OR) and 95% confidence intervals (CI) of the association between TC and LDL-C and speciated arsenic.

Speciated Arsenics	Median (IQR)-μg/L	80th versus 20th Percentile	OR (95% CI) of TC	OR (95% CI) of LDL-C
Arsenous acid	0.80 (0.34, 0.85)	0.85/0.08	0.99 (0.81, 1.21)	1.20 (0.98, 1.46)
Arsenic acid	0.62 (0.56, 0.71)	0.71/0.56	1.35 (1.02, 1.79) *	1.22 (0.91, 1.62)
Arsenobetaine	1.29 (0.82, 6.58)	9.37/0.82	0.99 (0.87, 1.13)	0.87 (0.74, 1.02)
Arsenocholine	0.32 (0.08, 0.42)	0.42/0.08	1.11 (0.84, 1.41)	1.15 (0.87, 1.52)
Dimethylarsinic	3.65 (2.11, 6.40)	7.39/1.87	1.09 (0.94, 1.28)	1.20 (1.03, 1.41) *
Monomethylarsonic	0.64 (0.56, 1.00)	1.13/0.36	1.14 (0.97, 1.34)	1.30 (1.11, 1.52) (***)

* *p* < 0.05; (***) *p* = 0.001.

**Table 6 nutrients-14-03993-t006:** Adjusted odds ratios (OR) and 95% confidence intervals (CI) of the association between TC and LDL-C and speciated arsenic.

Speciated Arsenics	OR (95% CI) of TC		OR (95% CI) of LDL-C	
	Model1 ^a^	Model2 ^b^	Model3 ^c^	Model4 ^d^
Arsenous acid	1.13 (0.91, 1.38)	1.17 (0.95, 1.44)	1.23 (1.00, 1.51) *	1.23 (1.01, 1.52) *
Arsenic acid	1.44 (1.08, 1.92) *	1.46 (1.10, 1.95) ^(^**^)^	1.24 (0.93, 1.66)	1.23 (0.92, 1.65)
Arsenobetaine	1.07 (0.93, 1.23)	1.07 (0.93, 1.23)	0.93 (0.79, 1.09)	0.91 (0.77, 1.07)
Arsenocholine	1.20 (0.90, 1.60)	1.19 (0.89, 1.59)	1.21 (0.91, 1.61)	1.20 (0.90, 1.59)
Dimethylarsinic	1.50 (1.25, 1.79) ***	1.47 (1.23, 1.76) ***	1.37 (1.15, 1.64) ^(^***^)^	1.34 (1.12, 1.61 ^(^***^)^
Monomethylarsonic	1.38 (1.16, 1.64) ***	1.36 (1.15, 1.62) ***	1.44 (1.22, 1.71) ***	1.43 (1.21, 1.70) ***

^a^ Model 1 is shown adds odds ratio (95% confidence interval); adjusted for creatine, BMI, and diabetes. ^b^ Model 2 is shown adds odds ratio (95% confidence interval); adjusted for creatine, BMI, diabetes, age, gender, and cotinine. ^c^ Model 3 is shown adds odds ratio (95% confidence interval); adjusted for creatine, BMI, diabetes, age, and drinking. ^d^ Model 4 is shown adds odds ratio (95% confidence interval); adjusted for creatine, BMI, diabetes, age, and marital status. *** *p* < 0.001; * *p* < 0.05; ^(^***^)^; *p* = 0.001. ^(^**^)^; *p* = 0.01.

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
