# Peer review of "Linking the Low-Density Lipoprotein-Cholesterol (LDL) Level to Arsenic Acid, Dimethylarsinic, and Monomethylarsonic: Results from a National Population-Based Study from the NHANES, 2003–2020"

_nutrients, 2022, doi:10.3390/nu14193993_

Round 1

Reviewer 1 Report

This is a potentially interesting paper to verify the association between low-density lipoprotein-cholesterol (LDL-C) level and urinary arsenic species levels using NHANES 2003-2020 data. This manuscript is similar to that published by Yue et al. in Int J Hgy Environ 2022-242-113950. This study only added the investigation of the association between urinary arsenic species and LDL cholesterol. However, there are major modifications that need to be made.

1.     TC, LDL-C, and HDL-C are all risk factors for cardiovascular disease. Therefore, the authors should fully explain in the Introduction why they only discuss TC and LDL-C, but not HDL-C. The authors should state the innovation of this study and how it differs from the paper in 2022 by Yue et al.

2.     For the determination of TC, LDL-C, HDL-C, total urinary arsenic and arsenic species, authors should describe the validity and reliability of all experiments and demonstrate the accuracy of the experiments. Urinary total arsenic or arsenic species concentrations measured in this study should be adjusted urine volume by urine specific gravity or urinary creatinine.

3.     The outcomes (dependent variables) of this study are TC and PDL-C, both of which are continuous variables and are only suitable for multivariate linear regression model analysis. If the dependent variables TC and LDL-C are divided into two categories: abnormal (1) and normal (0), the multivariate logistic regression model analysis is applied. The significance and purpose of the authors' distinction between TC and LDL-C as 70th vs. 30th or 80th vs. 20th should be explained.

4.     Was the independent variable urinary arsenic concentration in Table 3 also divided into 70th vs. 30th and 80th vs. 20th? Why not use a continuous variable for urinary total arsenic or arsenic species concentrations to see how much each unit increase affects TC or LDL-C? Or analyze the effect on TC or LDL-C by urinary arsenic concentration or arsenic species to increase one tertile or one quartile?

5.     Was the grouping of TC or LDL-C in Table 4, 70th vs 30th? Or 80th vs 20th?

6.     Was the independent variable (arsenic species) in Table 5, 80th vs. 20th? Or is it a continuous variable?

7.     Was the grouping of TC or LDL-C in Table 6, 70th vs. 30th? Or 80th vs. 20th? Was the independent variable (arsenic species), 80th vs. 20th? 70th vs. 30th? Or a continuous variable?

8.     The logic of statistical analysis of data should be clear. It is recommended to analyze the effect on TC or LDL-C with increasing urinary total arsenic concentration or arsenic species concentration (continuously or per increasing tertile or quartile).

9.     The control for confounding factors should be risk factors for outcome, and these factors in turn are related to the independent variable (urinary total arsenic or arsenic species concentration). There are significant factors in Table 1, but in the subsequent analysis (Table 2 to Table 6), not adjustments for all significant factors were made. The author should explain why the selective adjustments were made?

10. The difference between this study and study in 2022 of Yue et al is the association between arsenic species concentration and TC and LDL-C. However, the associations or mechanisms of arsenic species on TC and LDL-C were not discussed in the Discussion section.

Author Response

September 2022, 8thMs. No.: nutrients-1892401Title: Linking the low-density lipoprotein-cholesterol (LDL) level to arsenic acid, dimethylarsinic, and monomethylarsonic: Results from a national population-based study from the NHANES, 2003–2020Corresponding Author: Prof. Ruixue HuangAuthors: Can Qu,

Respond letter

 Dear Dr. : Hope anything for you all are doing well so far.Thank you for giving us a chance to make revision and resubmit. It is also very much appreciated cordially for reviewers’ encouraging and constructive advices and big encouragement on our manuscript entitled “Linking the low-density lipoprotein-cholesterol (LDL) level to arsenic acid, dimethylarsinic, and monomethylarsonic: Results from a national population-based study from the NHANES, 2003–2020 (No. nutrients-1892401)”.

Carefully, we studied your instructions as well as reviewers’ comments and suggestions. We have addressed each of concerns in the manuscript and a summary of the responses is outlined below. All authors have read and approved the final revised version, agreed to order of authorship for this manuscript, as well as agreed to the revised submission. We have checked the authors’ names and confirm they are all in the format that they wish to see appear in print. Attached please find the revised version, which we are pleased to resubmit for your kind consideration. In addition, according to the editor’s suggestion, the main text should not be less than 4000 words, so we have expanded the content of the paper to more than 4000 words.

We think our work is very valuable because our work shows that arsenic exposure at a higher dose in the population has a positive effect on the increase of TC and LDL-C concentration, thus providing basic information for the prevention and control of cardiovascular and cerebrovascular diseases, metabolic syndrome and other diseases. It would help scientists further study the mechanism or help public to get more understanding on association between arsenic exposure and serum lipids or lipid protein metabolism. Also, it will attract more and more studies on solving metabolic diseases to improve human health. This is why we like to do this work.

We would like to express our great appreciation to you and reviewers for comments and suggestions on our paper, meanwhile, I hope that you find our response satisfactory and the manuscript have chance to be acceptable.

I am looking forward to hearing from you.

Thank you and best regards.

Yours sincerely,

Corresponding author: Name: Dr. Ruixue Huang

E-mail: huangruixue@csu.edu.cn

  Response to editor Response to reviewers

Reviewer 1

Comments to the Author

This is a potentially interesting paper to verify the association between low-density lipoprotein-cholesterol (LDL-C) level and urinary arsenic species levels using NHANES 2003-2020 data. This manuscript is similar to that published by Yue et al. in Int J Hgy Environ 2022-242-113950. This study only added the investigation of the association between urinary arsenic species and LDL cholesterol. However, there are major modifications that need to be made.

1.TC, LDL-C, and HDL-C are all risk factors for cardiovascular disease. Therefore, the authors should fully explain in the Introduction why they only discuss TC and LDL-C, but not HDL-C. The authors should state the innovation of this study and how it differs from the paper in 2022 by Yue et al.

Response: Thank you for your comments. We have explained the reason why only discuss TC and LDL-C in this paper.

1In order to use the existing analysis results to extract more instructive and clear conclusions, we have focused more on TC and LDL-C with urinary arsenic concentrations correlation study. We chose to use the NHANES database to mainly explore the correlation between LDL-C and urinary arsenic concentration, and used the analysis of the correlation between TC and urinary arsenic concentration to support the previous findings. This is a validation and supplement of previous studies, and can make this analysis more targeted.

(2) There are five main differences between the paper of Yue and us:

1)Firstly, our paper differs from the paper of Yue et al. study subjects, whose article focused on 12-17 years old adolescents, and our study is for adults over 20 years old, which is not only a supplement to the study subjects of previous article, but also a study covering a wider age span of the population.

2)Secondly, as mentioned above, we explored the relationship between TC, LDL-C and urine arsenic concentration, while Yue's article focused more on the relationship between TC, HDL-C and urine arsenic concentration.

3)Thirdly, we have enrolled 6633 participants and Yue’s article included 1777 objects.

4) Fourthly, our research time range is from 2003 to March 2020, and Yue's article time range varied from 2009 to 2016.

5) In addition, the statistical methods of the two papers are different, we divided the urine arsenic into multiple concentration segments. First using the linear relationship analysis to explore the correlation between the urine arsenic quartiles with the TC and LDL-C concentrations, and then compare the TC and LDL concentrations of urine arsenic 30% vs. 70%, 20% vs.80%, obtaining ORs of those with high urine arsenic concentrations to the dangerous threshold TC or LDL-C concentrations. The main analysis methods used in Yue’ article is linear regression, which is an effective and worth learning methods, too.

2.For the determination of TC, LDL-C, HDL-C, total urinary arsenic and arsenic species, authors should describe the validity and reliability of all experiments and demonstrate the accuracy of the experiments. Urinary total arsenic or arsenic species concentrations measured in this study should be adjusted urine volume by urine specific gravity or urinary creatinine.

(1)Thank you for your advice, we have added the TCLDL-C and urinary arsenic and arsenic species in methods, and adjusted the creatine in analysis model.

 According The NHANES Lab Procedures Manual, the specific methods of quality assurance in sample collection and specific detection methods are listed in 2.3. The Urine Arsenic Assessment and 2.4 The Total cholesterol (TC) and low-density lipoprotein cholesterol (LDL-C) sections. Urine arsenic, urinary speciated arsenic, TC, LDL-C are measured by their own high-specific detection methods, strict information entry and quality control are carried out before sample using, and some operations are repeated 2-3 times during detection to ensure the reliability of experimental results.

(2) Urinary total arsenic or arsenic species concentrations measured in this study were adjusted by incorporating urinary creatinine into regression analysis models.

3.The outcomes (dependent variables) of this study are TC and PDL-C, both of which are continuous variables and are only suitable for multivariate linear regression model analysis. If the dependent variables TC and LDL-C are divided into two categories: abnormal (1) and normal (0), the multivariate logistic regression model analysis is applied. The significance and purpose of the authors' distinction between TC and LDL-C as 70th vs. 30th or 80th vs. 20th should be explained.

(1)   Thank you for your comments, we have explained the significance and purpose of grouping in statistical analysis according your suggestion.

We classified TC and LDL-C as health risks and healthy ranges according to previous literature and research reports, while urinary arsenic concentration was divided into 70th vs. 30th and 80th vs.20th to run a logistic regression analysis. Comparing the extremely high range of urinary arsenic distribution with the extreme low range of urinary arsenic distribution in the population, and exploring that people in which stage of urinary arsenic distribution are more likely to reach the risk range of TC or LDL-C, then resulting in more direct and credible clinically instructive results.

  1. Was the independent variable urinary arsenic concentration in Table 3 also divided into 70th vs. 30th and 80th vs. 20th? Why not use a continuous variable for urinary total arsenic or arsenic species concentrations to see how much each unit increase affects TC or LDL-C? Or analyze the effect on TC or LDL-C by urinary arsenic concentration or arsenic species to increase one tertile or one quartile?

(1) Thank you for your advice, we have analyzed the effect on TC or LDL-C by urinary arsenic concentration to increase one quartile in table 2.

(2) Only urinary arsenic concentrations were divided into 70th vs. 30th and 80th vs. 20th, the remaining covariates were still divided by yes, no, or specific definitions. Because the distribution of urine arsenic is right skewed and the change in each unit of urine arsenic concentration is not obvious enough to have a significant effect on TC or LDL-C. It is possible that the urine arsenic concentration must reach a certain threshold before it can be linked to TC and LDL-C, and analyzing the correlation directly using continuous variables may mask the effect of true changes in urine arsenic concentrations on TC and LDL-C concentrations. This analysis is complementary to the method of converting the urine arsenic concentration to quartiles and then performing multiple linear regression analysis.

(3) In order to understand the changes in TC and LDL-C concentrations caused by the change of urine arsenic concentration with different levels, we have divided the urine arsenic concentration into quartiles to explore the changes of TC and LDL-C brought about by the increase of each aliquottable 2. In order to avoid the complicated structure of the article, this multiple linear regression analysis is only a preliminary exploration of the correlation between total arsenic and lipoprotein, and the multiple linear correlation analysis of urine arsenic species was not reflected in the article.

  1. Was the grouping of TC or LDL-C in Table 4, 70th vs 30th? Or 80th vs 20th?

Thank you for your kind question, we have also added explanations in the text introduction section of table 4. TC and LDL-C are still divided into normal and abnormal, but the urine arsenic concentration is divided into quartiles, with the lowest concentration as the reference group, calculating closer to TC and LDL-C outliers ORs between higher concentration level with the lowest urine arsenic concentration.

  1. Was the independent variable (arsenic species) in Table 5, 80th vs. 20th? Or is it a continuous variable?

Thank you for your problem. we have added explanations in the text introduction section of table 5. We divided arsenic species concentration into 80th and 20th, which is a categorical variable, TC and LDL-C are still divided into normal and abnormal, then obtaining crude ORs with abnormal TC and LDL-C in arsenic species concentrations of 80th vs.20th without correlated factors adjusting.

  1. Was the grouping of TC or LDL-C in Table 6, 70th vs. 30th? Or 80th vs. 20th? Was the independent variable (arsenic species), 80th vs. 20th? 70th vs. 30th? Or a continuous variable?

Thank you for your meaningful question, we have added explanations in the text introduction section of table 6. TC and LDL-C are still divided into normal and abnormal, but each urine arsenic species concentration is grouped by 80th and 20th, supplementary explanations are provided in the corresponding explanation of table 6.

  1. The logic of statistical analysis of data should be clear. It is recommended to analyze the effect on TC or LDL-C with increasing urinary total arsenic concentration or arsenic species concentration (continuously or per increasing tertile or quartile).

Thanks for your advice, we have cleared analysis process and I would like to explain the data analysis of this article:

(1) The urine arsenic concentration is divided into quartiles, and the TC and LDL-C concentration changes brought about by the change of each urine arsenic concentration quartile are explored by multiple linear regression analysis, and the β coefficient is obtained. (table 2)

(2) Set the TC, LDL outlier dividing line and divide it into abnormal or dangerous range, normal range, and compare 70th vs. 30th concentration and 80th vs. 20th urine arsenic concentrations to obtain ORs closer to the abnormal range of TC, LDL-C. (table 3)

(3) TC and LDL-C are still divided into normal and abnormal, and the urine arsenic concentration is divided into quartiles. ORs closer to the abnormal range of TC, LDL-C complying with the urine arsenic concentration gradient contrast to the lowest total arsenic quartile were calculated respectively, and perform trend testing. (table 4)

(4) TC and LDL-C are still divided into normal and abnormal, comparing the 80th vs. 20th arsenic concentration of each species, obtain the OR without correlated factor adjustments, and then the OR after multi-model adjustment. (table 56)

  1. The control for confounding factors should be risk factors for outcome, and these factors in turn are related to the independent variable (urinary total arsenic or arsenic species concentration). There are significant factors in Table 1, but in the subsequent analysis (Table 2 to Table 6), not adjustments for all significant factors were made. The author should explain why the selective adjustments were made?

Thanks for your comments, we have added relevant explanatory notes to the data analysis section. First of all, the confounding factors that might be widely explored in the early stage are analyzed to facilitate the inclusion of model control confusion. The statistically significant factors in Table 1 have been included in the model of initial analysis, but after several adjustments, the goodness of fit of the model is not well, and the OR value of the included item such as race in the model is not statistically significant, so the covariate with no statistical significance is discarded in the final conformed model to ensure the validity of the whole regression model.

10.The difference between this study and study in 2022 of Yue et al is the association between arsenic species concentration and TC and LDL-C. However, the associations or mechanisms of arsenic species on TC and LDL-C were not discussed in the Discussion section.

Thanks for your advice, we have added the discussion of association between arsenic species and TC or LDL-C in discussion section of this paper. Among urine speciated arsenic analysis, the dimethylarsinic and monomethylarsonic concentration rising are more likely to related to the TC or LDL-C abnormal.

Reviewer: 2

Comments to the Author

A retrospective study indicating that arsenic exposure increases LDL and TC levels. Previous studies have also shown that arsenic exposure can affect lipid metabolism by reducing serum HDL levels and increasing serum LDL levels. A major issue is that (as authors outline) that there is no data regarding patients' medication. An addition regarding arsenic resources would be useful. Last, a short reference to arsenic exposure association with diabetes, hypertension, metabolic syndrome would be interesting.

Thank you very much for your comments, we have edited the paper according to your request.

(1)   The participants' treatment and medication status had a certain effect on the concentration of TC and LDL-C, and the reason why this factor was not considered in this article is that the number of people with this information, such as whether they take lipid-lowering drugs, is not enough to carry out the analysis, which is only 1/4 of the current number of participants included in the study .If this is used as an exclusion criterion, it will produce a selection bias ,thereby reducing the validity of the analysis.

(2)   A description of the arsenic resources has been added in the discussion section.

(3)   A brief introduction to the association of increased arsenic exposure to diabetes, hypertension, and metabolic syndrome has been stated in discussion.

Reviewer 2 Report

A retrospective study indicating that arsenic exposure increases LDL and TC levels. Previous studies have also shown that arsenic exposure can affect lipid metabolism by reducing serum HDL levels and increasing serum LDL levels. A major issue is that (as authors outline) that there is no data regarding patients' medication. An addition regarding arsenic resources would be useful. Last, a short reference to arsenic exposure association with diabetes, hypertension, metabolic syndrome would be interesting.

Author Response

(The authors gave the same response as above.)

Round 2

Reviewer 1 Report

In order to make this paper more readable, the following matters need to be corrected.

1.     The author explained that the difference from Yue's paper is clear, but this paper is only a similar study. To show the ingenuity of this study is it possible to propose that the association between arsenic exposure and TC or and LDL-C may vary by age or period?

2.     I may not have made it clear that I am not asking the authors to elaborate on the methods for the determination of TC, LDL-C and urinary arsenic and arsenic species. The method only needs a brief description and a cited reference. I think the reliability and validity of the methods should be described, such as coefficient of variation (CV), recovery rate (sample spiking) or standard reference material (SRM).

3.     In line 194-199 "Comparing the extremely high range of urinary arsenic distribution with the extreme low range of urinary arsenic distribution in the population, and exploring that people in which stage of urinary arsenic distribution are more likely to reach the risk range of TC or LDL-C, the odds ratios (OR) of TC, LDL-C and urine arsenic concentration were calculated at the 20th with 80th percentiles, and 30th with 70th percentiles. "

In line 260-261 " The results showed that people with 80% of total urine arsenic levels were more likely to reach the dangerous threshold of TC and LDL-C than those with 20%, 70 % versus 30 % as well."

Because TC and LDL-C are continuous variables, they become categorical variables to estimate OR, and the cut point needs to be well-founded (preferably citing references) and reasonable, so the cut point should be clearly defined. Authors should clearly define risk thresholds for TC and LDL-C when they begin to analyze the association between total urinary arsenic and arsenic species concentrations and TC and LDL-C.

4.     "Among urine speciated arsenic analysis, the dimethylarsinic and monomethylarsonic concentration rising are more likely to be related to the TC or LDL-C abnormal." This result was a finding of this study and should be discussed by the authors.

Author Response

September 2022, 15thNutrients-1892401Title: Linking the low-density lipoprotein-cholesterol (LDL) level to arsenic acid, dimethylarsinic, and monomethylarsonic: Results from a national population-based study from the NHANES, 2003–2020Corresponding Author: Prof. Ruixue HuangAuthors: Can Qu,

Respond letter

 Dear Dr. : Hope anything for you all are doing well so far.Thank you for giving us a chance to make revision and resubmit again. It is also very much appreciated cordially for reviewers’ encouraging and constructive advices and big encouragement on our manuscript entitled “Linking the low-density lipoprotein-cholesterol (LDL) level to arsenic acid, dimethylarsinic, and monomethylarsonic: Results from a national population-based study from the NHANES, 2003–2020 (Nutrients-1892401)”.

Carefully, we studied your instructions as well as reviewers’ comments and suggestions. We have addressed each of concerns in the manuscript and a summary of the responses is outlined below. All authors have read and approved the final revised version, agreed to order of authorship for this manuscript, as well as agreed to the revised submission. We have checked the authors’ names and confirm they are all in the format that they wish to see appear in print. Attached please find the revised version, which we are pleased to resubmit for your kind consideration.

We think our work is very valuable because our work shows that arsenic exposure at a higher dose in the population has a positive effect on the increase of TC and LDL-C concentration, thus providing basic information for the prevention and control of cardiovascular and cerebrovascular diseases, metabolic syndrome and other diseases. It would help scientists further study the mechanism or help public to get more understanding on association between arsenic exposure and serum lipids or lipid protein metabolism. Also, it will attract more and more studies on solving metabolic diseases to improve human health. This is why we like to do this work.

We would like to express our great appreciation to you and reviewers for comments and suggestions on our paper, meanwhile, I hope that you find our response satisfactory and the manuscript have chance to be acceptable.

I am looking forward to hearing from you.

Thank you and best regards.

Yours sincerely,

Corresponding author: Name: Dr. Ruixue Huang

E-mail: huangruixue@csu.edu.cn

  Response to editor Response to reviewers

Reviewer 1

Comments to the Author

In order to make this paper more readable, the following matters need to be corrected.

  1. The author explained that the difference from Yue's paper is clear, but this paper is only a similar study. To show the ingenuity of this study is it possible to propose that the association between arsenic exposure and TC or and LDL-C may vary by age or period?

Response: Thank you for your comments. We have added to the description of the results in forest chart(fig.3) the depictions of the relationship between urinary arsenic concentration and TC and LDL-C in different age groups and discussed it in the discussion section.

  1. I may not have made it clear that I am not asking the authors to elaborate on the methods for the determination of TC, LDL-C and urinary arsenic and arsenic species. The method only needs a brief description and a cited reference. I think the reliability and validity of the methods should be described, such as coefficient of variation (CV), recovery rate (sample spiking) or standard reference material (SRM).

Thank you for your comments. We have added the corresponding CV range description to the methods description of urine arsenic measurement and TC, LDL-C measurement, respectively.

  1. In line 194-199 "Comparing the extremely high range of urinary arsenic distribution with the extreme low range of urinary arsenic distribution in the population, and exploring that people in which stage of urinary arsenic distribution are more likely to reach the risk range of TC or LDL-C, the odds ratios (OR) of TC, LDL-C and urine arsenic concentration were calculated at the 20th with 80th percentiles, and 30th with 70th percentiles. "

In line 260-261 " The results showed that people with 80% of total urine arsenic levels were more likely to reach the dangerous threshold of TC and LDL-C than those with 20%, 70 % versus 30 % as well."

Because TC and LDL-C are continuous variables, they become categorical variables to estimate OR, and the cut point needs to be well-founded (preferably citing references) and reasonable, so the cut point should be clearly defined. Authors should clearly define risk thresholds for TC and LDL-C when they begin to analyze the association between total urinary arsenic and arsenic species concentrations and TC and LDL-C.

Thank you for your comments. We have clearly defined the risk thresholds for TC and LDL-C in Participant characteristics partline 227-233.

4."Among urine speciated arsenic analysis, the dimethylarsinic and monomethylarsonic concentration rising are more likely to be related to the TC or LDL-C abnormal." This result was a finding of this study and should be discussed by the authors.

(1) Thank you for your comments, we have discussed the finding of the correlation between dimethylarsinicmonomethylarsonic concentration and TCLDL-C in discussion.

Reviewer 2 Report

AUTHORS HAVE RESPONDED TO REVIEWERS' COMMENTS  PROPERLY

Author Response

(The authors gave the same response as above.)
